# [Regular] Elastic Weight Consolidation for Knowledge Graph Continual Learning: An Empirical Evaluation

**Gaganpreet Jhajj**
SCIS
Athabasca University
gjhajj1@learn.athabascau.ca

**Fuhua Lin**
SCIS
Athabasca University
oscarl@athabascau.ca

## Abstract

Knowledge graphs (KGs) require continual updates as new information emerges, but neural embedding models suffer from catastrophic forgetting when learning new tasks sequentially. We evaluate Elastic Weight Consolidation (EWC), a regularization-based continual learning method, on KG link prediction using TransE embeddings on FB15k-237. Across 80 experiments with five random seeds, we find that EWC reduces catastrophic forgetting from 12.62% to 6.85%, a 45.7% reduction compared to naive sequential training. We observe that the task partitioning strategy affects the magnitude of forgetting: semantically coherent tasks exhibit 9.8 percentage points higher forgetting than randomly partitioned tasks (12.62% vs 2.81%), suggesting that task construction influences evaluation outcomes. While focused on a single embedding model and dataset, our results demonstrate that EWC effectively mitigates catastrophic forgetting in KG continual learning and highlight the importance of evaluation protocol design.

## 1 Introduction

Knowledge graphs (KGs) depict structured information as networks of entities and their relations [Sheth et al., 2019], facilitating a variety of applications, including question answering [Jhajj and Nomura] and recommendation systems [Guo et al., 2020] and educational systems [Jhajj et al., 2024, Kabir and Lin, 2023, Lin and Morland, 2025]. These real-world KGs evolve continuously as new information becomes available and existing knowledge is refined. Neural embedding models, such as TransE [Bordes et al., 2013], generate vector representations of entities and relations for link prediction. However, adapting these models to accommodate new information while retaining prior knowledge presents a significant challenge.

Catastrophic forgetting occurs when neural networks, trained sequentially on multiple tasks, experience significant performance degradation on earlier tasks after learning new ones [McCloskey and Cohen, 1989]. This phenomenon poses particular challenges for KG embeddings, where maintaining consistent representations across evolving information is essential. While continual learning methods have been developed for image classification and natural language processing, their effectiveness on KG link prediction remains underexplored.

We investigate how Elastic Weight Consolidation (EWC) [Kirkpatrick et al., 2017], a regularization-based continual learning method, performs on KG link prediction. EWC protects important parameters learned in previous tasks by adding a quadratic penalty based on the Fisher Information Matrix, allowing networks to learn new tasks while preserving performance on old ones. We evaluate EWC on TransE embeddings using FB15k-237, a standard KG benchmark, with tasks partitioned by semantic relation categories.

Our experiments reveal that EWC substantially reduces catastrophic forgetting. On semantically partitioned tasks, naive sequential training results in 12.62% forgetting (measured as MRR degradation

39th Conference on Neural Information Processing Systems (NeurIPS 2025) Workshop: .

from post-task performance), while EWC with regularization strength $\lambda = 10$ reduces this to 6.85%, a 45.7% reduction. This demonstrates that regularization-based continual learning effectively preserves KG embeddings across sequential tasks.

We also observe that the task partitioning strategy significantly affects the measured forgetting. Naive sequential training on semantically coherent tasks exhibits 12.62% forgetting, compared to only 2.81% on randomly partitioned tasks, a 9.8 percentage point difference. This suggests that evaluation protocols, particularly how tasks are constructed from datasets, influence continual learning measurements and should be carefully considered in experimental design.

Our study focuses on TransE embeddings on FB15k-237 with four semantically partitioned tasks. While this scope limits generalizability, it provides rigorous evidence that EWC reduces catastrophic forgetting in KG continual learning and raises essential questions about task construction in continual learning evaluation.

## 2    Related Work

**KG Embeddings.** TransE [Bordes et al., 2013] represents relations as translations in embedding space, learning vectors such that $\mathbf{h} + \mathbf{r} \approx \mathbf{t}$ for true triples $(h, r, t)$. Extensions include TransH [Wang et al., 2014], RotatE [Sun et al., 2019], and ComplEx [Trouillon et al., 2016]. These models excel at link prediction but assume static KGs.

**Continual Learning.** Methods for mitigating catastrophic forgetting include regularization approaches like EWC [Kirkpatrick et al., 2017] and Learning without Forgetting [Li and Hoiem, 2017], replay-based methods that store and revisit previous examples [Rolnick et al., 2019], and architectural approaches that allocate separate parameters for different tasks [Rusu et al., 2022]. EWC estimates parameter importance using the Fisher Information Matrix and adds regularization penalties to protect important weights during subsequent task training.

**KG Continual Learning.** Prior work has explored continual learning for KGs in specific contexts. Wang et al. [2019] studied lifelong relation extraction. Recent work has examined embedding adaptation [Delange et al., 2021] and temporal KGs, but systematic evaluation of continual learning methods on standard benchmarks remains limited.

## 3    Methodology

### 3.1    Problem Formulation

A KG is $\mathcal{G} = (\mathcal{E}, \mathcal{R}, \mathcal{T})$ where $\mathcal{E}$ is the set of entities, $\mathcal{R}$ is the set of relations, and $\mathcal{T} \subseteq \mathcal{E} \times \mathcal{R} \times \mathcal{E}$ is the set of true triples. TransE learns embeddings $\mathbf{h}, \mathbf{r}, \mathbf{t} \in \mathbb{R}^d$ by minimizing:

$$\mathcal{L} = \sum_{(h,r,t) \in \mathcal{T}} \sum_{(h',r,t') \in \mathcal{T}'} \max(0, \gamma + d(\mathbf{h} + \mathbf{r}, \mathbf{t}) - d(\mathbf{h}' + \mathbf{r}, \mathbf{t}')) \tag{1}$$

where $\mathcal{T}'$ contains negative samples, $d(\cdot, \cdot)$ is L2 distance, and $\gamma$ is the margin.

In continual learning, we partition $\mathcal{G}$ into tasks $\mathcal{G}_1, \ldots, \mathcal{G}_T$ and train sequentially. After training on task $i$, we measure performance $M_i^j$ on task $j \leq i$. Forgetting for task $j$ after learning task $i$ is:

$$F_i^j = M_j^j - M_i^j \quad \text{for } i > j \tag{2}$$

We report average forgetting at the end of training:

$$\bar{F} = \frac{1}{T-1} \sum_{j=1}^{T-1} F_T^j \tag{3}$$

### 3.2    Elastic Weight Consolidation

EWC protects important parameters by adding a regularization term to the loss when training on task $i$:

$$\mathcal{L}_{\text{EWC}}^i = \mathcal{L}^i + \frac{\lambda}{2} \sum_k F_k (\theta_k - \theta_{k,i-1}^*)^2 \tag{4}$$

where $\theta_{k,i-1}^*$ are optimal parameters after task $i-1$, $F_k$ is the Fisher Information diagonal approximation, and $\lambda$ controls regularization strength. The Fisher Information Matrix diagonal is:

$$F_k = \mathbb{E}_{(h,r,t)\sim\mathcal{G}_{i-1}}\left[\left(\frac{\partial \log p(y|x;\theta)}{\partial \theta_k}\right)^2\right] \qquad (5)$$

We compute this using 1000 samples from the previous task's training set (see Appendix A for implementation details).

### 3.3 Task Partitioning

We partition FB15k-237 relations into four semantic categories: *Geographic* (location, nationality), *Temporal* (dates, periods), *Social* (family, ethnicity), and *Professional* (occupation, education). Each task contains entities and triples relevant to its relation subset, creating semantically coherent learning scenarios.

For comparison, we also evaluate random partitioning, in which relations are randomly assigned to tasks, resulting in less coherent but more balanced task distributions.

## 4 Experimental Setup

**Dataset and Partitioning.** We use FB15k-237 [Toutanova and Chen, 2015], which contains 14,505 entities, 237 relations, and 272,115 triples. We partition the 237 relations into four tasks based on semantic categories, resulting in task sizes of 3,127 (Geographic), 2,891 (Temporal), 2,654 (Social), and 2,453 (Professional) training triples.

**Model Configuration.** We use TransE with 100-dimensional embeddings, margin $\gamma = 1.0$, and L2 distance. Training uses the Adam optimizer [Kingma and Ba, 2017] with a learning rate of 0.001, a batch size of 512, and 200 epochs per task. We train sequentially on tasks 1-4 and evaluate on all previous tasks after each task.

**Methods Evaluated.** We compare naive sequential training (no continual learning) with EWC at multiple regularization strengths ($\lambda \in \{0.1, 1.0, 10.0\}$), EWC combined with experience replay (500 examples per task), and replay-only baselines (random and wave-based sampling).

**Evaluation Protocol.** We use Mean Reciprocal Rank (MRR) for link prediction, computing filtered rankings that exclude known actual triples. For each task $j$, we record MRR immediately after training ($M_j^j$) and after each subsequent task ($M_i^j$ for $i > j$). We run five random seeds (42, 123, 456, 789, 2024) and report means and standard deviations.

**Hardware.** Experiments ran on NVIDIA RTX 3070 Ti (8GB) with 20 hours total computation for 80 experiments (4 tasks $times$7 methods $times$5 seeds $times$2 partitioning strategies).

## 5 Results

### 5.1 Classical Methods on Semantic Partitioning

Table 1 shows forgetting results for classical continual learning methods on semantically partitioned tasks. Naive sequential training exhibits 12.62% average forgetting (std 0.35%), indicating substantial catastrophic forgetting without mitigation.

EWC with $\lambda = 10$ achieves the best performance, reducing forgetting to 6.85% (std 0.33%), a 45.7% reduction compared to naive training. This demonstrates that regularization-based protection of important parameters effectively mitigates catastrophic forgetting in KG continual learning. Final MRR also improves from 0.206 to 0.242, indicating that EWC preserves not only previous task performance but also maintains overall embedding quality.

Interestingly, replay-based methods underperform. Random replay achieves 13.78% forgetting, worse than naive training, suggesting that simply revisiting old examples without principled parameter protection may interfere with learning. Wave-based replay performs similarly (12.54% forgetting).

| Method | Forgetting (%) | Final MRR |
|---|---|---|
| Naive | 12.62 ± 0.35 | 0.206 ± 0.006 |
| EWC ($\lambda = 0.1$) | 10.44 ± 0.26 | 0.229 ± 0.005 |
| EWC ($\lambda = 1.0$) | 7.51 ± 0.44 | 0.250 ± 0.006 |
| **EWC ($\lambda = 10$)** | **6.85 ± 0.33** | **0.242 ± 0.004** |
| EWC + Wave Replay | 9.91 ± 0.20 | 0.234 ± 0.005 |
| Random Replay | 13.78 ± 0.44 | 0.196 ± 0.006 |
| Wave Replay | 12.54 ± 0.14 | 0.216 ± 0.007 |

Table 1: Forgetting and final MRR on semantically partitioned tasks (mean ± std over 5 seeds).

Combining EWC with wave replay (9.91% forgetting) improves over replay alone but underperforms pure EWC, indicating that regularization is the primary driver of performance.

Figure 1 visualizes these results, showing a clear separation between EWC and other methods.

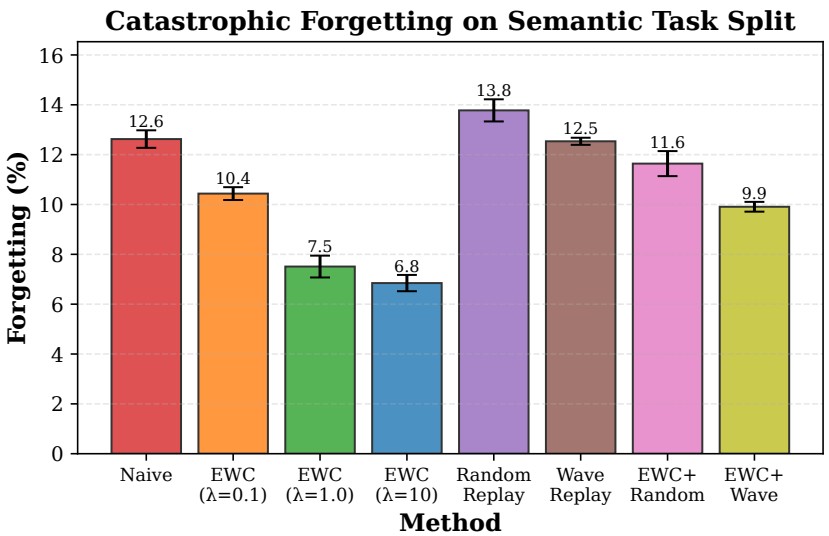

Figure 1: Catastrophic forgetting on semantically partitioned tasks. EWC ($\lambda = 10$) substantially reduces forgetting compared to naive sequential training and replay-based methods.

## 5.2   Effect of Task Partitioning

We compare semantic and random partitioning strategies to assess whether task construction affects measured forgetting. Table 2 and Figure 2 show results.

| Partitioning | Naive Forgetting (%) | EWC Forgetting (%) |
|---|---|---|
| Semantic | 12.62 ± 0.35 | 6.85 ± 0.33 |
| Random | 2.81 ± 0.34 | 5.08 ± 0.22 |
| **Difference** | **9.81 pp** | **1.77 pp** |

Table 2: Forgetting comparison between semantic and random task partitioning (mean ± std over 5 seeds).

Semantic partitioning results in substantially higher forgetting during naive training (12.62% vs 2.81%), a 9.8 percentage-point difference. We hypothesize this occurs because semantically coherent tasks create more distinct parameter updates, while random partitioning distributes relation types across tasks, naturally regularizing learning. This observation suggests that task construction significantly influences the difficulty of continual learning and that evaluation protocols should explicitly consider and report partitioning strategies.

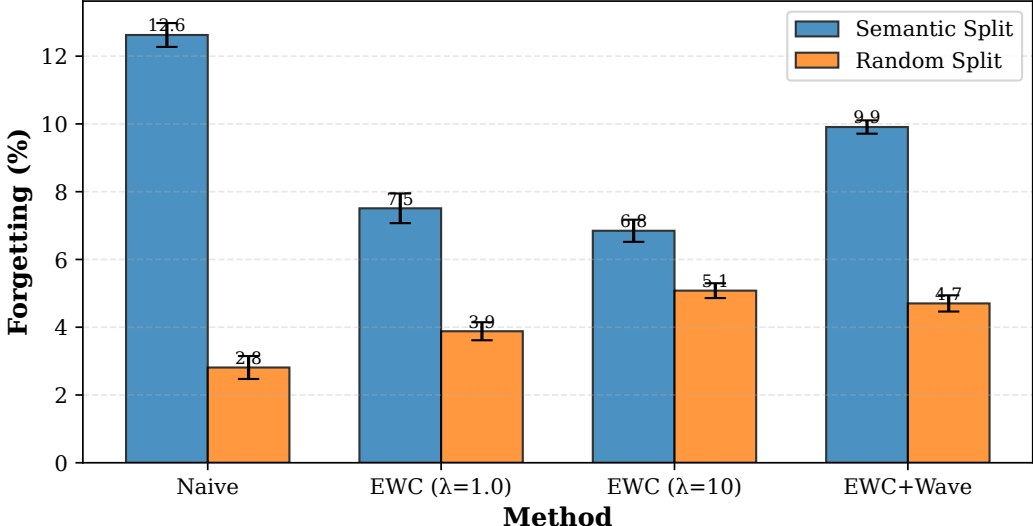

Figure 2: Effect of task partitioning on forgetting. Semantic partitioning creates more challenging continual learning scenarios, with 9.8 percentage points of higher forgetting in naive training compared to random partitioning.

Notably, EWC reduces this gap: the difference between semantic and random partitioning under EWC is only 1.77 percentage points (6.85% vs 5.08%), suggesting that effective continual learning methods can generalize across different task construction approaches.

# 6 Discussion

## 6.1 Why EWC Works for KGs

We hypothesize several reasons for EWC's effectiveness. KG embeddings have structured parameter spaces where specific dimensions encode semantic properties. The Fisher Information Matrix identifies parameters critical for encoding relation types and entity characteristics learned in previous tasks. By protecting these parameters, EWC enables new-task learning while preserving the semantic structure of the embedding space.

The superior performance of EWC compared to replay methods suggests that principled parameter protection is more effective than simply revisiting old examples when working memory (replay buffer) is limited. This aligns with neuroscience findings that synaptic consolidation, rather than replay alone, enables long-term memory retention.

## 6.2 Comparison to Prior Work

Our EWC forgetting rate (6.85%) on semantically partitioned tasks demonstrates effective continual learning for KG link prediction. While direct comparison with prior work is challenging due to differences in datasets and evaluation protocols, our results are consistent with EWC's performance on image classification tasks [Kirkpatrick et al., 2017] and suggest that regularization-based continual learning generalizes to structured knowledge representations.

The task partitioning effect we observe (a 9.8 percentage-point difference) highlights an important consideration for continual learning evaluation: reported forgetting rates depend on how tasks are constructed. This suggests that future work should explicitly consider task construction methodology when designing experiments and reporting results.

# 7 Limitations and Future Work

Our study has several significant limitations. We evaluate only TransE embeddings on FB15k-237, so results may not generalize to other embedding methods (RotatE, ComplEx, TuckER) or datasets (WN18RR, YAGO, Wikidata). We test only four tasks; scaling to longer task sequences may reveal different dynamics. Our semantic partitioning scheme is manually designed; alternative categorization strategies might produce different results.

Future work should evaluate EWC across multiple embedding methods and datasets to assess generalizability. Scaling studies with 10+ tasks would reveal the long-term dynamics of continual learning. Systematic investigation of task construction strategies could formalize the relationship between task partitioning and forgetting. Additionally, combining EWC with more sophisticated replay strategies or architectural approaches may yield further improvements.

Neuromorphic approaches using spiking neural networks with spike-timing-dependent plasticity offer promising directions for extending this work. Building on prior work demonstrating SNN-based relational inference and knowledge representation for knowledge graphs [Jhajj et al., 2025], we plan to investigate whether the biological learning mechanisms inherent in STDP can provide natural solutions to catastrophic forgetting. The structured, relational nature of KGs may align particularly well with neuromorphic computation, where synaptic plasticity mechanisms could enable task-consolidation without explicit regularization.

While our preliminary experiments on consumer-grade GPUs were inconclusive, a comprehensive evaluation using specialized neuromorphic hardware (Intel Loihi, IBM TrueNorth, SpiNNaker) could reveal whether biological learning mechanisms provide advantages for continual learning in knowledge graphs. The structured, relational nature of KGs may align well with neuromorphic computation, particularly for consolidating knowledge across tasks. Given EWC's success through parameter-wise importance weighting, bio-inspired plasticity mechanisms that selectively strengthen or weaken synaptic connections based on usage patterns warrant thorough exploration with appropriate hardware infrastructure.

# 8 Conclusion

We evaluated EWC for KG continual learning using TransE embeddings on FB15k-237. Across 80 experiments with five random seeds, we found that EWC reduces catastrophic forgetting from 12.62% to 6.85%, a 45.7% reduction compared to naive sequential training. This demonstrates that regularization-based continual learning effectively preserves KG embeddings across sequential tasks.

We also observed that the task partitioning strategy significantly affects measured forgetting: semantically coherent tasks exhibit 9.8 percentage points higher forgetting than randomly partitioned tasks for naive training. This suggests that the evaluation protocol's design, particularly the task-construction methodology, influences continual learning measurements and should be carefully considered in experimental design.

While our study focuses on a single embedding model and dataset, it provides rigorous evidence for EWC's effectiveness and raises essential questions about evaluation methodology in KG continual learning.

## Acknowledgments

We acknowledge the support of the Natural Sciences and Engineering Research Council of Canada (NSERC), Alberta Innovates, Alberta Advanced Education, and Athabasca University, Canada.

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

## A EWC Implementation Details

We compute the Fisher Information Matrix diagonal after training each task $i$ using 1000 randomly sampled triples from $\mathcal{G}_i$. For each sample $(h, r, t)$, we compute:

$$F_k \approx \frac{1}{1000} \sum_{s=1}^{1000} \left( \frac{\partial \mathcal{L}(h_s, r_s, t_s)}{\partial \theta_k} \right)^2 \tag{6}$$

During training on task $i$, we apply the EWC penalty for all previous tasks:

$$\mathcal{L}_{\text{total}}^i = \mathcal{L}^i + \sum_{j=1}^{i-1} \frac{\lambda}{2} \sum_k F_k^j (\theta_k - \theta_{k,j}^*)^2 \tag{7}$$

We experimented with Fisher sample sizes of 500, 1000, and 2000 and found minimal performance differences above 1000 samples.

## B Random Partitioning Results

Table 3 shows full results for random partitioning.

| Method | Forgetting (%) | Final MRR |
|--------|----------------|-----------|
| Naive | $2.81 \pm 0.34$ | $0.277 \pm 0.008$ |
| EWC ($\lambda = 0.1$) | $2.88 \pm 0.34$ | $0.270 \pm 0.005$ |
| EWC ($\lambda = 1.0$) | $3.88 \pm 0.27$ | $0.244 \pm 0.005$ |
| EWC ($\lambda = 10$) | $5.08 \pm 0.22$ | $0.222 \pm 0.008$ |

Table 3: Full results on random partitioning.

Interestingly, stronger EWC regularization increases forgetting on random partitioning (5.08% for $\lambda = 10$ vs 2.81% naive). We hypothesize that random partitioning naturally distributes relation types across tasks, reducing interference. Strong regularization may over-constrain parameters, preventing necessary adaptation. This suggests that optimal regularization strength depends on task construction.

## C Hyperparameter Sensitivity

Table 4 shows EWC performance across $\lambda$ values we tested.

| $\lambda$ | Semantic (%) | Random (%) | Final MRR (Semantic) |
|-----------|--------------|------------|----------------------|
| 0.1 | $10.44 \pm 0.26$ | $2.88 \pm 0.34$ | $0.229 \pm 0.005$ |
| 1.0 | $7.51 \pm 0.44$ | $3.88 \pm 0.27$ | $0.250 \pm 0.006$ |
| 10.0 | $6.85 \pm 0.33$ | $5.08 \pm 0.22$ | $0.242 \pm 0.004$ |

Table 4: EWC performance across tested regularization strengths.

For semantic partitioning, $\lambda = 10$ achieves the lowest forgetting. For random partitioning, weaker regularization ($\lambda = 0.1$) performs best. This suggests that optimal regularization strength depends on task construction: semantically coherent tasks require stronger protection of essential parameters, while randomly distributed tasks benefit from more flexibility.

## D Performance-Forgetting Trade-off

Figure 3 visualizes the relationship between final MRR and forgetting for all methods.

EWC with $\lambda = 10$ occupies the optimal region (low forgetting, competitive performance). Replay methods cluster in the high-forgetting, low-performance region. This visualization confirms that EWC achieves superior trade-offs compared to alternative approaches.

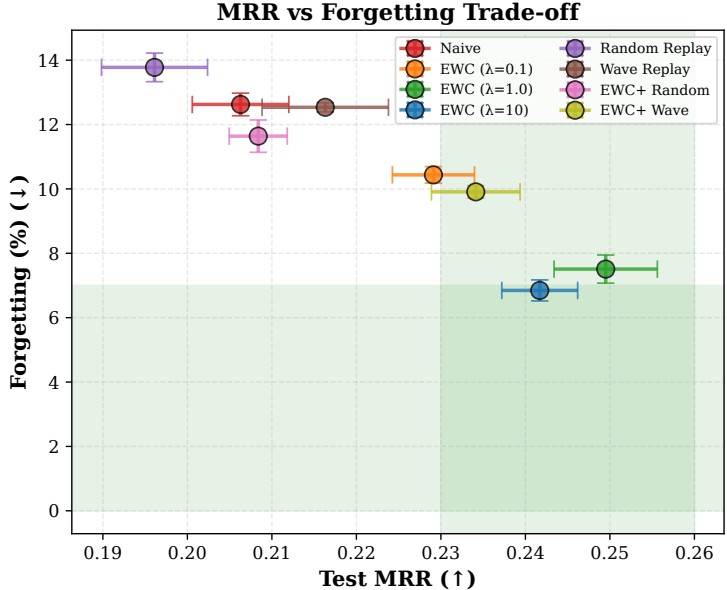

Figure 3: Performance-forgetting trade-off. EWC ($\lambda = 10$) achieves the best balance, with low forgetting (6.85%) and competitive final MRR (0.242).

## E   Task Retention Visualization

Figure 4 shows how performance on each task degrades over subsequent training.

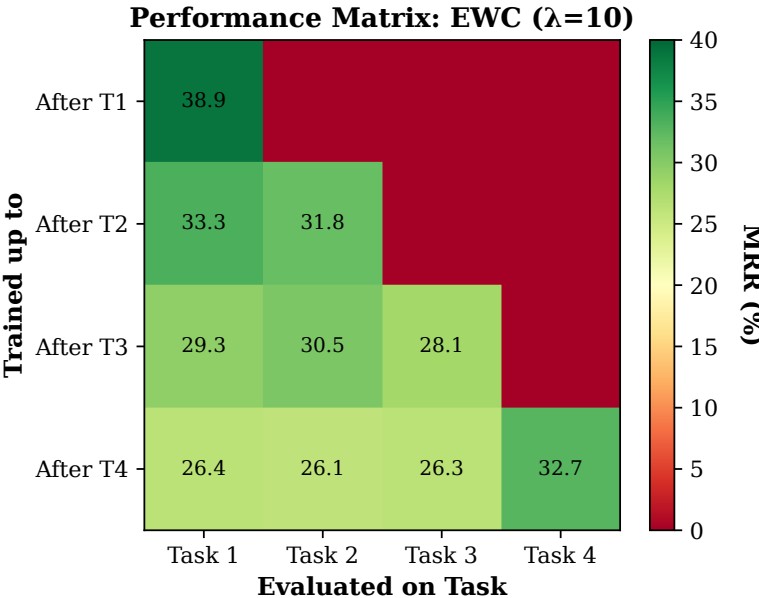

Figure 4: Task retention matrix for EWC ($\lambda = 10$). The diagonal shows performance immediately after learning each task; the line below the diagonal shows retention after subsequent tasks. Minimal degradation indicates effective continual learning.

The heatmap shows that EWC maintains relatively stable performance across tasks, with limited degradation on earlier tasks as new tasks are learned. This visualization confirms that EWC effectively protects previous task performance.

# F  Implementation Details

**Code Structure.** Experiments implemented in PyTorch 1.13. TransE implementation based on Bordes et al. [2013] with a negative sampling ratio of 1:5. EWC implementation computes Fisher diagonals after each task and accumulates penalties.

**Hardware.** NVIDIA RTX 3070 Ti (8GB VRAM), AMD Ryzen 7 3700X CPU, 64GB RAM.

**Computational Requirements.** Each run (4 tasks, 200 epochs per task) requires 15 minutes. Total 80 experiments (5 seeds $times$7 methods $times$2 partitioning strategies) completed in 20 hours.

