# OpenReview forum: "Elastic Weight Consolidation for Knowledge Graph Continual Learning: An Empirical Evaluation"
_NeurIPS.cc/2025/Workshop_Mexico_City/NORA — NeurIPS 2025 Workshop NORA Poster_

### Official Review · Reviewer_uEQf · 2025-11-01
**Continual Learning for TransE: Limited Novelty and Insufficient Analysis of Prior Work**

**Rating:** 3
**Confidence:** 4

**Review:**

This paper presents a study on applying continual learning to TransE, a knowledge graph embedding model. The authors employ Elastic Weight Consolidation (EWC) to achieve continual learning and evaluate their approach on the FB15k-237 dataset. Experimental results show that the proposed method outperforms naive sequential learning in terms of performance.

However, the paper does not sufficiently analyze or position itself within the context of prior work on continual learning for knowledge graph embeddings [1,2]. Moreover, the proposed approach, applying EWC to continual learning—lacks methodological novelty, as EWC is a well-established technique in the continual learning literature. Although the authors acknowledge as a limitation that experiments were conducted only on the TransE model and the FB15k-237 dataset, this restriction further limits the generalizability of the results.

[1] Daruna et al., Continual Learning of Knowledge Graph Embeddings, arXiv:2101.05850
[2] Zhao et al., Rethinking Continual Knowledge Graph Embedding: Benchmarks and Analysis, SIGIR 2025

---

### Official Review · Reviewer_HTKE · 2025-11-05
**This paper conducts an empirical study on Elastic Weight Consolidation for knowledge graph continual learning**

**Rating:** 5
**Confidence:** 4

**Review:**

The paper scope is quite limited. It only considers EWC as the base model. But I am wondering if the results can be expanded to other models? Note that there are a lot of new works on knowledge graph continual learning. Another suggestion is that the authors may consider the unlearning task, which means to forget some outdated or unwanted knowledge. This is also a practical question in knowledge graph continual learning.

---

### Official Review · Reviewer_TQHF · 2025-11-05

**Rating:** 5
**Confidence:** 3

**Review:**

Paper Summary:

The paper evaluates Elastic Weight Consolidation (EWC), a regularization-based continual learning method, for knowledge graph link prediction using TransE embeddings on FB15k-237. Across 80 experiments, EWC reduces catastrophic forgetting by 45.7% compared to naive sequential training. The evaluation results show that how tasks are partitioned matters: semantically coherent task splits lead to dramatically more forgetting (12.62%) than randomly partitioned tasks (2.81%), highlighting the need for careful evaluation protocol design in KG continual learning.

Pros:
1. The paper is clearly written and the presentation is easy to follow.
2. The empirical analyses are comprehensive.

Cons:
1. The novelty appears somewhat limited, as the work mainly evaluates an existing method on an existing task without introducing new methodological elements.
2. The findings would be more convincing and generalizable if the authors evaluated a wider range of embedding models to verify whether the observed trends hold across architectures.
3. The motivation for choosing **this** specific method and **this** specific task is not sufficiently articulated. It would be helpful to explain why this combination is particularly meaningful or non-trivial.
4. The relevance of the work to the workshop theme is unclear. The paper does not seem to involve any agent-related components, which makes its fit with the workshop somewhat questionable.

---

### Official Review · Reviewer_qq5f · 2025-11-06
**Elastic Weight Consolidation for Knowledge Graph Continual Learning**

**Rating:** 4
**Confidence:** 3

**Review:**

The paper highlights the usage of Elastic Weight Consolidation methodology to reduce the catastrophic forgetting while training Knowledge graph on new tasks. The paper highlights that EWC protects important parameters by adding a regularization term to the loss when training on new tasks. The paper also compares random partitioning vs semantic partitioning and highlights that semantic partitioning suffers more from catastrophic forgetting, which is more akin to real world scenario.

As highlighted in the limitations section, the research needs to be more thorough for drawing concrete conclusions for Knowledge Graph Continual Learning:
* The Related Work section should involve more current research on the state-of-the-art Knowledge Graph Continual Learning methodologies and compare the proposed methodology against it. There have been several papers that use Bayesian or low rank adaptations for continually training the Knowledge graphs.
* The methodology is only compared against one dataset and 4 tasks. The experiment should be performed on multiple datasets and varied tasks to highlight its importance.
* Some ablation studies would be helpful to highlight that the effectiveness of the loss function.

---

### Official Review · Reviewer_7TkW · 2025-11-06
**Early research on EWC for knowledge graphs**

**Rating:** 6
**Confidence:** 4

**Review:**

**Criteria**

- Novelty: The paper reuses an existing concept, EWC, in a different space, KGs
- Clarity: The paper reads well.
- Reproducibility: The authors use a well-known KG completion algorithm and dataset; however, they fail to share some details
- Ethical Compliance: The paper doesn't comment on ethical concerns, societal impacts, or biases, and there are none I could think of

**Comments**

This paper explores the idea of using Elastic Weight Consolidation as a regularisation method to address catastrophic forgetting in knowledge graph embedding learning. The continual learning task is designed by dividing the KG into semantically related groups of relations. The TransE model is used on the FB15k-237 dataset to evaluate this, which yields promising results.

The idea is interesting and attractive, given the real-world applications of KG embedding learning. However, because the approach was tested on a single model (TransE, which is one of the early models), I can't stop thinking that the results are too early to extrapolate. The decision to adopt that model is not discussed, and one can ask why not other models, such as ComplEx, that generally provide better results. There are many missing details as well that affect the results of the model training, e.g., the corruption method used to generate negative samples is never mentioned, and this has been identified as critical in previous work. The authors should also consider sharing their partition of the FB15k-237 dataset for reproducibility and future comparative experiments.

When discussing the results (Section 5), the authors refer to the good results with $\lambda = 10$ as a demonstration. This is by no means a formal demonstration, but rather an observation from the experiment design. The same experiment should be validated with other KG embedding algorithms and datasets to provide more confidence in the efficacy of what's being proposed. Moreover, the results show a tendency to decrease forgetting when the $\lambda$ increases; however, there are only 3 values for $\lambda$, and nothing higher than 10 was evaluated. Why? Since we don't know the answer to this question, I'd be cautious with certain claims as they may be invalidated with further experiments.

Finally, looking at the details provided in the appendix, I can see that the Fisher Information Matrix for each task is computed for each triple, and FB15k-237 has 592213 triples! Sampling is applied, but there is no discussion on the representativeness of the samples. Further discussion on the complexity of the proposed method is required, and suggestions for potential optimisations.

**Small comments**
1. Section 1, second paragraph should provide references for the sentence "While continual learning methods have been developed for image classification and natural language processing, ..."
2. Section 3.1, add "defined as" -> "A KG is [defined as] \mathcal{G} = ..."
3. Section 4, towards the end, the number experiments says `times`, it should be `\times` -> $\times$. Same issue found in Appendix F.

---

### Official Review · Reviewer_iDHE · 2025-11-07

**Rating:** 7
**Confidence:** 3

**Review:**

This paper presents an empirical evaluation of Elastic Weight Consolidation applied to knowledge graph link prediction. It provides comparative analysis of task partitioning strategies, demonstrating that semantically coherent task splits induce significantly higher forgetting compared to random splits.

Novelty:
The systematic application and rigorous evaluation presented in this paper within the specific context of KG embedding is valuable.  The empirical evidence showing how heavily task construction influences established is novel.

Clarity:
The paper is well-written and clearly structured. The problem statement is precise and figures are used properly to illustrate the main point regarding EWC performance.

Reproducibility:
The paper provides sufficient detail for reproduction including hyperparams and implementaiton details.

Weaknesses:
the evaluation is limited to a single embedding model and dataset with only four tasks. Generalizability to more complex models remains unexplored.

---

### Official Review · Reviewer_EYLr · 2025-11-07
**Comments by the Reviewer**

**Rating:** 6
**Confidence:** 3

**Review:**

**Summary**

This paper investigates the problem of continual learning in knowledge graph embeddings. The authors argue that knowledge graphs (KGs) need to be continuously updated as new information emerges, yet neural embedding models tend to suffer from catastrophic forgetting when learning new tasks sequentially. To address this issue, the authors adopt the Elastic Weight Consolidation (EWC) regularization-based continual learning method and evaluate its effectiveness on the FB15k-237 dataset using the TransE embedding technique for link prediction.
Through 80 experiments with five random seeds, the results show that EWC significantly reduces the forgetting rate compared with naive sequential training. Furthermore, the authors observe that task partitioning strategies influence the extent of forgetting: semantically coherent tasks exhibit lower forgetting rates than randomly partitioned ones, suggesting that the way tasks are constructed affects evaluation outcomes.
Although the study focuses on a single embedding model and dataset, the results demonstrate that EWC effectively mitigates forgetting in continual learning for knowledge graphs.

**Strengths**

1. The authors address an important problem in knowledge graph research — how to alleviate catastrophic forgetting in continual learning.

2. The paper is well-motivated, clearly identifying the challenge and proposing Elastic Weight Consolidation (EWC), a regularization-based continual learning approach, as a potential solution.

3. The manuscript is well written and easy to follow.

**Weaknesses**

1. The authors only conducted experiments on a single dataset, making it unclear how robust the proposed method is across different datasets or settings.

2. Moreover, given that large language models (LLMs) also suffer from catastrophic forgetting, it would be interesting to see whether the proposed approach could be generalized or extended to LLMs in future work.

---

### Official Review · Reviewer_YdxR · 2025-11-07
**The evaluation method is outdated**

**Rating:** 5
**Confidence:** 4

**Review:**

The paper presents an empirical study on applying Elastic Weight Consolidation (EWC) to address catastrophic forgetting in knowledge graph (KG) continual learning. Using TransE on FB15k-237, the authors report that EWC reduces forgetting by around 45% compared to naive sequential training. They also examine how task partitioning affects forgetting, finding that semantically coherent splits lead to higher forgetting rates.

While the paper is clearly written and the experiments are systematically executed, its scientific contribution is minimal. EWC is a well-established method published in 2017, and directly applying it to an old model (TransE, 2013) and dataset (FB15k-237, 2017) offers limited novelty. There are no comparisons with other continual learning methods, which makes the evaluation incomplete.